# Non-C_2_-Symmetric *Bis*-Benzimidazolium Salt Applied in the Synthesis of Sterically Hindered Biaryls

**DOI:** 10.3390/molecules26216703

**Published:** 2021-11-05

**Authors:** Yen-Hsin Chen, Shu-Jyun Huang, Tung-Yu Hsu, Pei-Yu Hung, Ting-Rong Wei, Dong-Sheng Lee, Ta-Jung Lu

**Affiliations:** Department of Chemistry, National Chung Hsing University, Taichung 40227, Taiwan; g107051122@mail.nchu.edu.tw (Y.-H.C.); g106051094@mail.nchu.edu.tw (S.-J.H.); g108051033@mail.nchu.edu.tw (T.-Y.H.); g105051016@mail.nchu.edu.tw (P.-Y.H.); g107051123@mail.nchu.edu.tw (T.-R.W.); tjlu@dragon.nchu.edu.tw (T.-J.L.)

**Keywords:** *bis*-benzimidazolium salt, sterically hindered biaryls, non-C_2_-symmetric

## Abstract

A novel non-C_2_-symmetric *bis*-benzimidazolium salt derived from (±)-valinol has been prepared by a simple and straightforward process in good yield. The structure of *bis*-benzimidazolium salt provided a bulky steric group on the ethylene bridge; which facilitates the catalytic efficacy in the C(*sp*^2^)–C(*sp*^2^) formation. Its catalytic activity in Suzuki–Miyaura cross-coupling reaction of unactivated aryl chlorides has been found to have high efficacy in 1 mol% Pd loading. This protocol demonstrated the potential on the synthesis of sterically hindered biaryls.

## 1. Introduction

Sterically hindered biaryls are present in many natural products and pharmaceuticals, such as vancomycin [1,2,3,4], steganacin [5], and michellamines B [6,7]. The Pd-catalyzed Suzuki–Miyaura cross-coupling reaction (SMC) is one of the most practical protocols for the construction of biaryls [8,9,10,11,12,13,14,15,16]. Numerous successful examples using a combination of phosphine ligands with palladium have been reported [9,10,12,13]. However, the drawbacks of the phosphine ligands are difficult preparation, air sensitivity, expensiveness, and toxicity. Hence, the development of Pd catalysts with phosphine-free ligands has been receiving great attention. Over the past three decades, the replacement of phosphine ligands with *N*-heterocyclic carbenes (NHCs) has been a good choice [8,11,15,16].

Aryl chlorides, which are less expensive and more diverse relative to aryl bromides and iodides, are noticeably challenging partners in SMC reaction due to their low C–Cl bond reactivity. The in situ-formed *bis*-NHC/Pd catalytic systems are easy to handle and provide two strong carbene–metal bonds, making them more stable than monodentate NHC/Pd species. In comparison to the broad study and application of palladium/*bis*-NHC systems in the SMC reaction of aryl bromides or activated chlorides [17,18,19,20,21,22,23,24,25,26,27,28,29,30,31], much less attention has been paid to unactivated aryl chlorides and the synthesis of sterically hindered biaryls. In 2014, a 1,2-cyclohexane-bridged *bis*-NHC palladium catalyst has been synthesized by Zhang and co-workers [26]. This catalyst was also applied in the Pd-catalyzed SMC reaction of 1-bromo-2-alkoxynaphthalene and 1-napthylboronic acid at 65 °C to afford 45–83% yield. Although the coupling between 1-chloro-2-methoxynaphthalene and 1-napthylboronic acid was also examined, a 52% yield was obtained in the presence of 3 mol% Pd loading. In 2018, Shi et al. reported the development of *bis*-NHC dipalladium complexes and their application in Suzuki–Miyaura cross-coupling reactions 1-bromonaphthalene with 1-naphthaleneboronic acid [28]. Moderate yields (32–52%) were obtained at 100 °C. In 2020, Zhang and Yu developed fine-tunable *bis*-NHC palladium catalysts [30]. These catalysts were also applied in the Pd-catalyzed SMC reaction between 1-bromo-2-alkoxynapthalene and 1-naphthylboronic acid at 40 °C in the presence of 2.5 mol% Pd loading to achieve 50–71% conversions. The grave challenges, sterically hindered aryl chlorides coupled with sterically hindered arylboronic acids with low Pd loading, still exist.

We recently reported an in situ-generated Pd(OAc)_2_/L·2HX catalyst for the Suzuki–Miyaura reaction of aryl bromides or aryl chlorides with arylboronic acids in good to excellent yields [32,33,34]. Motivated by these results we continued our efforts to develop an efficient in situ–generated Pd(OAc)_2_/L·2HX catalytic system to catalyze the synthesis of biaryls. Herein, we describe the preparation of a new non-C_2_-symmetric *bis*-benzimidazolium salt (Figure 1) and its application in the coupling between sterically hindered aryl chlorides and arylboronic acids.

## 2. Results

### 2.1. Synthesis and Characterization of the Bis-Benzimidazolium Salts ***3***

Chiral valinol is often used to prepare chiral ligands, such as oxazolines [35], which are employed in asymmetric catalysis with excellent efficiency. (±)-Valinol is chosen as the starting material because it has a non-C_2_-symmetric ethylene skeleton, which can be used as a linker of non-C_2_-symmetric *bis*-benzimidazolium salt. In addition, the isopropyl group on the ethylene bridge could act as a bulky steric group which will facilitate the reductive elimination step in the catalytic cycle. The *bis*-benzimidazole **2** was synthesized by a simple and straightforward process from valinol. The *bis*-benzimidazolium salt **1** was obtained by the combination of **2** and two equivalents of benzyl bromide in acetonitrile at reflux in 91% yield (Figure 1). The new salt was air- and moisture-stable both in the solid state and in solution. It was characterized by ^1^H- and ^13^C-NMR. The two benzimidazolium proton signals exhibit as sharp singlets at δ 12.51 and 12.25 ppm in the ^1^H-NMR spectrum, and two corresponding carbon resonances appear as a typical singlet in the ^1^H-decoupled mode at δ 143.7 and 142.3 ppm in the ^13^C-NMR spectrum.

### 2.2. The Suzuki–Miyaura Cross-Coupling Reaction

Continuing our previous studies on the application of in situ-formed catalyst to SMC reaction [32,33,34], the SMC reaction of 4-chloroanisole **3a** with phenylboronic acid **4a** with 1.0 mol% Pd loading was chosen to study the optimized reaction conditions (Table 1). Solvents were evaluated firstly (entries 1–5), and 1,4-dioxane showed that **5aa** had a good GC yield (entry 1, 87%). Secondly, the Pd/**1** ratio from 1:0.5 to 1:3 (entries 1 and 6–8) were examined, and the ratio of 1:3 was found to give the highest yield (87%) (entry 1). The base is usually an important factor in this reaction (entries 1 and 9–15). Among commonly used bases, K_3_PO_4_·H_2_O was found to be the best base. Finally, various metal sources were investigated in SMC reaction (entries 1 and 16–21), and Pd(dba)_2_ was the metal source of choice for this reaction (entry 21, 83% isolated yield). A control reaction was also performed in the absence of *bis*-benzimidazolium salt **1** (entry 22), which showed that no starting material was converted to biaryl **5aa**.

After the optimized reaction conditions were secured, the scope of SMC reaction of aryl chlorides was studied (Table 2). Unactivated aryl chlorides **3a** and **3b** were successfully coupled with phenylboronic acid in the presence of 1.0 mol% Pd loading with moderate to good yields (entries 1 and 2). Functionalized aryl chlorides were successfully coupled with phenylboronic acid with good to excellent yields (entries 3–6, 87–99%) except **5fa** (entry 7, 63%). *o*-Monosubstituted aryl chlorides could couple with phenylboronic acid to achieve corresponding biaryls in 66–85% yields (entries 8–11). Our results clearly show that the catalytic system formed by this non-C_2_-symmetric *bis*-benzimidazolium salt, possessing high steric hindrance, and the Pd source demonstrated excellent catalytic capacity. It catalyzes the C(*sp*^2^)–C(*sp*^2^) bond formation between deacitvated aryl chlorides and phenylboronic acid under low Pd loading (1 mol%). In addition, this catalyst has superior tolerance to a wide range of functional groups.

In order to further extend the capacity of the catalytic system, the synthesis of sterically hindered biaryls was scrutinized (Table 3). Di-*ortho*-substituted aryl chlorides, such as 2-chloro-1-methyl-3-nitrobenzene, 2-chloro-3-methoxy benzaldehyde, and 2-chloro-1,3-dimethylbenzene, were coupled with phenylboronic acid to afford the corresponding products in 46–60% yields (entries 1–3). Di-*ortho*-substituted biaryls (**5hb**, **5ib**, **5pb**, and **5kb**) were derived from 1-naphthylboronic acid and aryl chlorides bearing *o*-functional groups, such as ketone, ester, and aldehyde in 20–92% yields (entries 4–7). Most of the tri*-ortho*-substituted biaryls (**5ob**, **5od**, **5nb**, **5ne**, and **5nf**) could be successfully generated in the presence of 1 mol% **1**/Pd(OAc)_2_ with 50–85% yields (entries 8–12). Unfortunately, the product **5nc** bearing tetra-*ortho*-substituents proved difficult to obtain (entry 13). So far, we have successfully demonstrated the synthesis of tri-*ortho*-substituted biaryls in the presence of 1 mol% in situ-formed a **1**/Pd(OAc)_2_ catalytic system.

## 3. Materials and Methods

### 3.1. General Methods

Unless otherwise stated, commercially available materials were received from Aldrich and Acros (New Taipei City, Taiwan) and used without further purification. Acetonitrile was distilled over calcium hydride prior to its use. Toluene, 1,4-dioxane, and *t*-BuOH were distilled over sodium prior to its use. Reactions were monitored using pre-coated silica gel 60 (F-254) plates. The products were purified by column chromatography (silica gel, 0.040–0.063 μm), eluting with *n*-hexane/ethyl acetate. ^1^H- and ^13^C-NMR spectra were recorded using an Agilent Mercury 400 spectrometer (Agilent Technologies, Inc., Santa Clara, CA, USA), with the *J*-values given in Hz. Chemical shifts (δ) were referenced to CDCl_3_ (δ = 7.26 ppm) in the ^1^H-NMR spectra and CDCl_3_ (δ = 77.0 ppm) in the ^13^C-NMR spectra. Copies of ^1^H- and ^13^C-NMR spectra of all compounds are provided as Appendix A. Melting points were determined using a Thermo 1001D digital melting point apparatus and are uncorrected. GC-FID was recorded using a Shimadzu GC-2014 spectrometer (Shimadzu Co., Kyoto, Japan) equipped with a capillary column (SPB^®^-5, 60 m × 0.25 mm × 0.25 μm). The conversion yields, GC yields, and ratios were determined using undecane as an internal standard. High-resolution mass spectra were recorded using a Finnigan/Thermo Quest MAT 95XL mass spectrometer (Finnigan MAT LCQ, San Jose, CA, USA) via either atmospheric-pressure chemical ionization (APCI) or electrospray ionization (ESI) methods.

### 3.2. Experimental Procedures and Spectral Data

#### 3.2.1. Synthesis of Non-C_2_-Symmetric *Bis*-Benzimidazolium Salt **1**

Procedure for 3-methyl-2-(2-nitrophenylamino)-butan-1-ol (**S1**) [36]. To a solution of 1-fluoro-2-nitrobenzene (1.10 equiv), valinol (0.10 g 0.97 mmol) and potassium carbonate (1.10 equiv) in DMSO (10 mL) was stirred for 2 h at 60 °C. The reaction mixture was cooled to room temperature and diluted with water (20 mL). The mixture was stirred for several minutes, then extracted with EtOAc (80 mL × 1). The organic layer was washed with saturated NaCl(aq) (20 mL × 3), dried over Na_2_SO_4_, and then filtered. The filtrate was concentrated under reduced pressure to afford crude product. The residual was purified by chromatography to obtain **S1** in a 66% yield (0.14 g). ^1^H-NMR (CDCl_3_, 400 MHz): δ 8.21 (br, 1H, NH), 8.17 (dd, *J* = 8.0, 1.2 Hz, 1H), 7.41 (t, *J* = 8.0 Hz, 1H), 6.99 (d, *J* = 8.0 Hz, 1H), 6.63 (t, *J* = 8.0 Hz, 1H), 3.85–3.81 (m, 1H), 3.77–3.72 (m, 1H), 3.64–3.60 (m, 1H), 2.06 (octect, *J* = 6.8 Hz, 1H), 1.66 (br, 1H, OH), 1.03 (d, *J* = 6.8 Hz, 3H), 1.02 (d, *J* = 6.8 Hz, 3H); ^13^C-NMR (CDCl_3_, 100 MHz): δ 146.1, 136.2, 131.9, 127.0, 115.3, 114.4, 63.1, 59.8, 29.7, 19.4, 18.3. Spectroscopic data was consistent with the literature [36].

Procedure for 2-[(2-aminophenyl)amino]-3-methylbutan-1-ol (**S****2**) [36]. Saturated NH_4_Cl(aq) (25 mL) was added to a mixture of Fe powder (10 equiv) and compound **S1** (0.12 g, 0.53 mmol) in EtOH (25 mL). After refluxing for 2 h, the resulting mixture was filtered. The filtrate was concentrated under reduced pressure. The residual was treated with H_2_O (20 mL) and EtOAc (30 mL). The aqueous layer was extracted with EtOAc (30 mL×3). The combined organic layers were dried over anhydrous Na_2_SO_4_ and then filtered. The filtrate was concentrated to afford black solid **S2** in 96% yield (0.20 g), which was subjected to the next reaction without any purification. ^1^H-NMR (CDCl_3_, 400 MHz): δ 6.81 (t, *J* = 7.6 Hz, 1H), 6.75 (d, *J* = 7.6 Hz, 1H), 6.72 (d, *J* = 7.6 Hz, 1H), 6.68 (t, *J* = 7.6 Hz, 1H), 3.78 (dd, *J* = 10.8, 4.0 Hz, 1H), 3.59 (dd, *J* = 10.8, 6.4 Hz, 1H), 3.37 (br, 2H, NH_2_), 3.31 (dd, *J* = 10.8, 6.4 Hz 1H), 1.97 (octect, *J* = 6.8 Hz, 1H), 1.12 (d, *J* = 6.8 Hz, 3H), 0.97 (d, *J* = 6.8 Hz, 3H); ^13^C-NMR (CDCl_3_, 100 MHz): δ 137.6, 134.1, 121.0, 118.7, 117.6, 113.3, 61.9, 60.4, 29.6, 19.3, 18.8. Spectroscopic data was consistent with the literature [36].

Procedure for 2-(1*H*-benzo[d]imidazol-1-yl-3-methylbutan-1-ol (**S3**) [36]. To a mixture of **S2** (0.20 g, 1.03 mmol), triethyl orthoformate (3.0 equiv) and sulfamic acid (0.05 equiv) in MeOH (20 mL) was stirred for 16 h at room temperature. After removal of MeOH, the resulting solution was treated with H_2_O (20 mL) and EtOAc (30 mL). The aqueous layer was extracted with EtOAc (30 mL × 3). The combined organic layers were dried over Na_2_SO_4_ and then filtered. The filtrate was concentrated on a rotary evaporator. The residual was purified by column chromatography to obtain **S3** in a 97% yield (0.20 g). ^1^H-NMR (CDCl_3_, 400 MHz): δ 7.93 (s, 1H), 7.65 (d, *J* = 7.6 Hz, 1H), 7.38 (d, *J* = 7.6 Hz, 1H), 7.24 (t, *J* = 7.6 Hz, 1H), 7.19 (t, *J* = 7.6 Hz, 1H), 4.17 (dd, *J* = 12.4, 7.6 Hz, 1H), 4.04–3.99 (m, 2H), 2.47–2.41 (m, 1H), 1.15 (d, *J* = 6.8 Hz, 3H), 0.76 (d, *J* = 6.8 Hz, 3H); ^13^C-NMR (CDCl_3_, 100 MHz): δ 141.9, 133.8, 122.7, 122.1, 119.7, 110.1, 65.2, 62.0, 29.3, 20.2, 19.7. Spectroscopic data was consistent with the literature [36].

Procedure for 2-(1*H*-benzo[d]imidazol-1-yl)-3-methylbutyl 4-methyl benzenesulfonate (**S4**). To a solution of **S3** (1.00 mmol) in pyridine (2 mL) was added *p*-toluenesulfonyl chloride (3.00 mmol) in one portion at 0 °C. The reaction mixture was stirred at room temperature overnight. The reaction mixture was diluted with EtOAc (30 mL) and then washed with saturated sodium bicarbonate (10 mL) and brine (10 mL). The organic layer was dried over Na_2_SO_4_ and filtered. The filtrate was concentrated on a rotary evaporator. The residual was purified by column chromatography to obtain **S4** in an 82% yield (0.72 g). Mp = 123–125 °C; ^1^H-NMR (CDCl_3_, 400 MHz): δ 7.82 (s, 1H), 7.75 (d, *J* = 8.0 Hz, 1H), 7.43 (d, *J* = 8.0 Hz, 1H), 7.26–7.17 (m, 3H), 7.07 (d, *J* = 8.0 Hz, 2H), 4.45 (dd, *J* = 10.8, 7.6 Hz, 1H), 4.36 (dd, *J* = 10.8, 3.2 Hz, 1H), 4.15–4.09 (m, 1H), 2.47–2.41 (m, 1H), 2.33 (s, 3H), 1.13 (d, *J* = 6.8 Hz, 3H), 0.74 (d, *J* = 6.8 Hz, 3H); ^13^C-NMR (CDCl_3_, 100 MHz): δ 145.1, 143.7, 141.9, 133.1, 131.5, 129.8, 127.4, 122.9, 122.2, 120.6, 109.8, 68.3, 61.8, 29.2, 21.6, 19.9, 19.5; MS–ESI (*m*/*z*) (relative intensity) 359.1 (M + H^+^, 100); HRMS–ESI (*m*/*z*) [M + H^+^]: calcd. for C_19_H_23_N_2_O_3_S: 359.1424, found: 359.1430.

Procedure for 1,1′-(3-methylbutane-1,2-diyl)*bis*(1*H*-benzo[d]imidazole) (**2**). The flask was charged with benzimidazole (1.10 equiv), KO*^t^*Bu (2.20 equiv). The solution of compound **S4** (0.22 g, 0.61 mmol) in DMF (6 mL) was added into the flask at room temperature at 0.5 mL/h. The reaction mixture was stirred at room temperature overnight and then treated with H_2_O (100 mL). The reaction mixture was extracted with EtOAc (3 × 100 mL). The combined organic layers were washed with brine, dried over Na_2_SO_4_, and filtered. The filtrate was concentrated on a rotary evaporator. The residual was purified by column chromatography to afford **2** (0.13 g, 68% yield). Mp = 190–195 °C; ^1^H-NMR (CDCl_3_, 400 MHz): δ 7.78 (d, *J* = 8.0 Hz, 1H), 7.77–7.13 (m, 1H), 7.55 (s, 1H), 7.26–7.13 (m, 7H), 4.80 (dd, *J* = 14.8, 3.6 Hz, 1H), 4.65 (dd, *J* = 14.8, 10.0 Hz, 1H), 4.27 (ddd, *J* = 14.8, 10.0, 3.6 Hz, 1H), 2.62–2.54 (m, 1H), 1.37 (d, *J* = 6.8 Hz, 3H), 0.85 (d, *J* = 6.8 Hz, 3H); ^13^C-NMR (CDCl_3_, 100 MHz): δ 143.0, 142.6, 141.8, 141.3, 132.3, 132.1, 122.4, 121.7, 121.6, 121.4, 119.7, 119.5, 109.2, 108.2, 62.5, 45.4, 29.5, 19.6, 18.6; MS–ESI (*m*/*z*) (relative intensity) 305.1 (M + H^+^, 100); HRMS–ESI (*m*/*z*) [M + H^+^]: calcd. for C_19_H_21_N_4_: 305.1761, found: 305.1770.

Procedure for 1,1′-(3-methylbutane-1,2-diyl)*bis*(3-benzyl-1*H*-benzo[d]imidazol-3-ium) dibromide (**1**). A mixture of 1,1′-(3-methylbutane-1,2-diyl)*bis*(1*H*-benzo[d]imidazole) (**2**) (0.7 g, 2.3 mmol) and benzyl bromide (5.1 mmol) was refluxed in acetonitrile (24.0 mL) for 13 h. The resulting mixture was filtered, and the residue was the *bis*-benzimidazolium salt **1** (1.3 g, 91%) as a white solid. ^1^H-NMR (CDCl_3_, 400 MHz): δ 12.51 (s, 1H), 12.25 (s, 1H), 8.92–8.88 (m, 1H), 8.80–8.78 (m, 1H), 7.49–7.20 (m, 16H,), 6.53 (s, 1H), 6.05–5.99 (m, 1H), 5.70–5.51 (m, 4H), 3.00–2.92 (m, 1H), 1.42 (d, *J* = 6.4 Hz, 3H), 0.97 (d, *J* = 6.8 Hz, 3H); ^13^C-NMR (CDCl_3_, 100 MHz): δ 143.7, 142.3, 132.0, 132.0, 129.8, 129.7, 129.4, 128.3, 128.2, 128.0, 127.8, 127.5, 127.4, 115.2, 114.9, 112.8, 64.7, 51.7, 49.3, 32.5, 19.8, 19.1; MS–ESI (*m*/*z*) (relative intensity) 243.4 (79), 395.4 (68), 485.4 (61), 565.2 (M + H^+^, 100), 567.3 (89); HRMS–ESI (*m*/*z*) [M−Br^−^] calcd. for C_33_H_34_N_4_Br: 565.1961, found 565.1944.

#### 3.2.2. General Procedures for Suzuki–Miyaura Cross-Coupling Reactions under N_2_

All manipulations were carried out under nitrogen using dried solvent. Pd(dba)_2_ (1 mol%), salt **1** (3 mol%) and 1,4-dioxane (3 mL), aryl chloride **3** (1.0 mmol), arylboronic acid **4** (1.5 mmol), and K_3_PO_4_·H_2_O (3.0 mmol) were charged to the Schlenk tube. After stirring for 24 h at 110 °C, the reaction was quenched by water (3.0 mL). The aqueous layer was extracted with EtOAc (3.0 mL × 3). The combined organic layers were dried over anhydrous Na_2_SO_4_ and then filtered. The solvent was evaporated under reduced pressure, and the corresponding product was purified by chromatography.

4-Methoxybiphenyl (**5aa**) [33]. ^1^H-NMR (CDCl_3_, 400 MHz): δ 7.57–7.51 (m, 4H), 7.42 (t, *J* = 7.6 Hz, 2H), 7.30 (t, *J* = 7.2 Hz, 1H), 7.98 (d, *J* = 8.8 Hz, 2H), 3.86 (s, 1H); ^13^C-NMR (CDCl_3_, 100 MHz): δ 159.1, 140.8, 133.7, 128.7, 128.1, 126.7, 126.6, 114.1, 55.3.

4-Methylbiphenyl (**5ba**) [34]. ^1^H-NMR (CDCl_3_, 400 MHz): δ 7.58 (d, *J* = 7.8 Hz, 2H), 7.50 (d, *J* = 8.0 Hz, 2H), 7.37 (t, *J* = 7.6 Hz, 2H), 7.32 (t, *J* = 7.6 Hz, 1H), 7.25 (d, *J* = 7.6 Hz, 2H), 2.40 (s, 1H); ^13^C-NMR (CDCl_3_, 100 MHz): δ 141.1, 138.3, 137.0, 129.5, 128.7, 127.0, 21.1.

4-Acetylbiphenyl (**5ca**) [33]. ^1^H-NMR (CDCl_3_, 400 MHz): δ 8.04 (d, *J* = 8.8 Hz, 2H), 7.69 (d, *J* = 8.8 Hz, 2H), 7.63 (d, *J* = 7.2 Hz, 2H), 7.48 (t, *J* = 7.2 Hz, 2H), 7.40 (t, *J* = 7.2 Hz, 1H), 2.65(s, 3H); ^13^C-NMR (CDCl_3_, 100 MHz): δ 197.7, 145.7, 139.8, 135.8, 128.9, 128.9, 128.2, 127.2, 127.2, 26.6.

4′-Phenylpropiophenone (**5da**) [33]. ^1^H-NMR (CDCl_3_, 400 MHz): δ 8.04 (d, *J* = 8.8 Hz, 2H), 7.69 (d, *J* = 8.8 Hz, 2H), 7.63 (d, *J* = 7.2 Hz, 2H), 7.47 (t, *J* = 7.2 Hz, 2H), 7.40 (t, *J* = 7.2 Hz, 1H), 3.45 (q, *J* = 7.2 Hz, 1H), 1.26 (t, *J* = 7.2 Hz, 3H); ^13^C-NMR (CDCl_3_, 100 MHz): δ 200.4, 145.5, 139.9, 135.6, 128.9, 128.5, 128.1, 127.23, 127.19, 31.8, 8.3.

4-Nitrobiphenyl (**5ea**) [34]. ^1^H-NMR (CDCl_3_, 400 MHz): δ 8.30 (d, *J* = 8.8 Hz, 2H), 7.74 (d, *J* = 8.8 Hz, 2H), 7.63 (d, *J* = 6.8 Hz, 2H), 7.52–7.45 (m, 3H); ^13^C-NMR (CDCl_3_, 100 MHz): δ 147.5, 147.0, 138.7, 129.1, 128.9, 127.7, 127.3, 124.0.

4-Cyanobiphenyl (**5fa**) [34]. ^1^H-NMR (CDCl_3_, 400 MHz): δ 7.73 (d, *J* = 8.8 Hz, 2H), 7.69 (d, *J* = 8.8 Hz, 2H), 7.59 (d, *J* = 7.2 Hz, 2H), 7.49 (t, *J* = 8.0 Hz, 2H), 7.43 (t, *J* = 7.2 Hz, 1H); ^13^C-NMR (CDCl_3_, 100 MHz): δ 145.6, 139.1, 132.5, 129.1, 128.6, 127.7, 127.2, 118.9, 110.9.

[1,1′-Biphenyl]-4-carbaldehyde (**5ga**) [33]. ^1^H-NMR (CDCl_3_, 400 MHz): δ 10.06 (s, 1H), 7.96 (d, *J* = 8.0 Hz, 2H), 7.76 (d, *J* = 8.4 Hz, 2H), 7.64 (d, *J* = 7.2 Hz, 2H), 7.49 (t, *J* = 7.2 Hz, 2H), 7.42 (t, *J* = 7.2 Hz, 1H); ^13^C-NMR (CDCl_3_, 100 MHz): δ 191.9, 147.2, 139.7, 135.2, 130.3, 129.0, 128.5, 127.7, 127.4.

2-Acetylbiphenyl (**5ha**) **[34]**. ^1^H-NMR (CDCl_3_, 400 MHz): δ 7.57–7.53 (m, 1H), 7.52–7.49 (m, 1H), 7.46–7.38 (m, 2H), 7.36–7.34 (m, 5H), 2.01 (s, 3H); ^13^C-NMR (CDCl_3_, 100 MHz): δ 204.9, 140.9, 140.7, 140.5, 130.7, 130.2, 128.8, 128.7, 127.9, 127.4, 30.4.

Methyl [1,1′-biphenyl]-2-carboxylate (**5ia**) [37]. ^1^H-NMR (CDCl_3_, 400 MHz): δ 7.83 (d, *J* = 8.0 Hz, 1H), 7.55–7.3 (m, 1H), 7.39–7.31 (m, 7H), 3.63 (s, 3H); ^13^C-NMR (CDCl_3_, 100 MHz): δ 169.1, 142.5, 141.1, 131.2, 130.9, 130.7, 129.8, 128.3, 128.0, 127.2, 127.1, 51.9.

2-Nitrobiphenyl (**5ja**) **[34]**. ^1^H-NMR (CDCl_3_, 400 MHz): δ 7.86 (d, *J* = 8.0 Hz, 1H) 7.62 (t, *J* = 7.2 Hz 1H), 7.51–7.40 (m, 5H), 7.34–7.32 (m, 2H); ^13^C-NMR (CDCl_3_, 100 MHz): δ 149.1, 137.2, 136.1, 132.1, 131.8, 128.5, 128.0, 127.7, 123.9.

2-Methyl-1,1′-biphenyl (**5ka**) **[34]**. ^1^H-NMR (CDCl_3_, 400 MHz): δ 7.42–7.39 (m, 2H), 7.36–7.31 (m, 3H), 7.27–7.26 (m, 1H), 7.25–7.23 (m, 3H), 2.28, (s, 1H); ^13^C-NMR (CDCl_3_, 100 MHz): δ 141.9, 135.2, 130.2, 129.7, 129.1, 128.0, 127.2, 126.7, 125.7, 20.4.

2-Methyl-6-nitro-1,1′-biphenyl (**5ma**) [38]. ^1^H-NMR (CDCl_3_, 400 MHz): δ 7.66 (d, *J* = 8.8 Hz, 1H), 7.49 (d, *J* = 7.6 Hz, 1H) 7.45–7.36 (m, 4H), 7.19 (d, *J* = 7.2 Hz, 2H), 2.14 (s, 3H); ^13^C-NMR (CDCl_3_, 100 MHz): δ 150.4, 139.1, 136.0, 135.3, 133.7, 128.5, 128.4, 127.9, 127.8, 121.0, 20.6.

6-Methoxy-[1,1′-biphenyl]-2-carbaldehyde (**5na**) [39]. ^1^H-NMR (CDCl_3_, 400 MHz): δ 9.74 (s, 1H), 7.63 (d, *J* = 7.6 Hz,1H), 7.48–7.42 (m, 4H), 7.33 (d, *J* = 6.4 Hz, 2H), 7.20 (d, *J* = 8.4 Hz, 1H), 3.79 (s, 1H); ^13^C-NMR (CDCl_3_, 100 MHz): δ 157.0, 135.4, 134.9, 133.1, 131.0, 128.7, 128.0, 127.9, 119.1, 115.9, 56.0.

2,6-Dimethyl-1,1′-biphenyl (**5oa**) [33]. ^1^H-NMR (CDCl_3_, 400 MHz): δ 7.43 (t, *J* = 7.4 Hz, 2H), 7.35 (d, *J* = 7.4 Hz, 1H), 7.18–7.11 (m, 5H), 2.04 (s, 6H); ^13^C-NMR (100 MHz, CDCl_3_): δ 141.8, 141.1, 136.0, 129.0, 128.4, 127.2, 127.0, 126.6, 20.8.

1-(2-(Naphthalen-1-yl)phenyl)ethan-1-one (**5hb**) [40]. ^1^H-NMR (CDCl_3_, 400 MHz): δ 7.92–7.89 (m, 2H), 7.75 (d, *J* = 6.8 Hz, 1H), 7.62 (d, *J* = 8.4 Hz, 1H), 7.60–7.49 (m, 4H), 7.43 (t, *J* = 7.6 Hz, 2H), 7.34 (d, *J* = 6.8 Hz, 1H), 1.79 (s, 3H); ^13^C-NMR (CDCl_3_, 100 MHz): δ 203.0, 141.2, 139.0, 138.6, 133.5, 131.8, 131.6, 130.8, 128.3, 128.3, 128.2, 127.7, 127.3, 126.5, 126.0, 125.6, 125.3.

Methyl 2-(naphthalen-1-yl)benzoate (**5ib**) [41]. ^1^H-NMR (CDCl_3_, 400 MHz): δ 8.03 (d, *J* = 7.6 Hz, 1H), 7.90–7.85 (m, 2H), 7.61 (t, *J* = 7.2 Hz, 1H), 7.54–7.31 (m, 7H), 3.36 (s, 3H,); ^13^C-NMR (CDCl_3_, 100 MHz): δ 167.9, 141.3, 139.6, 133.2, 132.0, 131.9, 131.6, 131.5, 130.0, 128.2), 127.5, 126.0, 125.9, 125.6, 125.5, 125.1, 51.7.

2-(Naphthalen-1-yl)benzaldehyde (**5pb**) [42]. ^1^H-NMR (CDCl_3_, 400 MHz): δ 79.63 (s, 1H), 8.12 (d, *J* = 7.6 Hz, 1H), 7.96–7.93 (m, 2H), 7.70 (t, *J* = 7.2 Hz, 1H), 7.61–7.41 (m, 7H); ^13^C-NMR (CDCl_3_, 100 MHz): δ 191.9, 144.1, 135.3, 134.7, 133.5, 133.3, 132.6, 131.6, 128.5, 128.3, 128.1, 127.0 (**C**6), 126.7, 126.1, 125.7, 124.9.

1-(*o*-Tolyl)naphthalene (**5kb**) [42]. ^1^H-NMR (CDCl_3_, 400 MHz): δ 7.91 (d, *J* = 8.4 Hz, 1H), 7.87 (d, *J* = 8.4 Hz, 1H) 7.55–7.51 (m, 1H), 7.50–7.45 (m, 2H), 7.40–7.33 (m, 5H), 7.32–7.28 (m, 1H), 2.03 (s, 3H); ^13^C-NMR (CDCl_3_, 100 MHz): δ 140.2, 139.8, 136.8, 133.5, 132.0, 130.3, 129.8, 128.2, 127.5, 127.4, 126.6, 126.1, 125.9, 125.7, 125.5, 125.4, 20.0.

1-(2,6-Dimethylphenyl)naphthalene (**5ob**) [43]. ^1^H-NMR (CDCl_3_, 400 MHz): δ 7.90 (d, *J* = 8.4 Hz, 1H), 7.86 (d, *J* = 8.4 Hz, 1H), 7.56–7.52 (m, 1H), 7.49–7.45 (m, 1H), 7.35–7.33 (m, 2H), 7.28–7.26 (m, 1H), 7.26–7.17 (m, 3H), 1.90 (s, 6H); ^13^C-NMR (CDCl_3_, 100 MHz): δ 139.6, 138.7, 137.0, 133.7, 131.7, 128.3, 127.3, 127.2, 127.2, 126.4, 126.0, 125.8, 125.7, 125.4, 20.4.

2,2′,6-Trimethyl-1,1′-biphenyl (**5od**) (=**5kc**) [44]. ^1^H-NMR (CDCl_3_, 400 MHz): δ 7.30–7.26 (m, 3H), 7.26–7.15 (m, 3H), 7.12–7.10 (m, 1H), 1.97 (s, 3H), 1.95 (s, 6H); ^13^C-NMR (CDCl_3_, 100 MHz): δ 140.5, 135.8, 135.6, 129.9, 129.8, 128.8, 127.2, 127.0, 126.9, 126.0.

3-Methoxy-2-(naphthalen-1-yl)benzaldehyde (**5nb**). ^1^H-NMR (CDCl_3_, 400 MHz): δ 9.47 (s, 1H), 7.95–7.91 (m, 2H), 7.71 (d, *J* = 8.8 Hz, 1H), 7.56 (t, *J* = 8.0 Hz, 2H), 7.51–7.48 (m, 1H), 7.41–7.38 (m, 3H), 7.28–7.26 (m, 1H), 3.70 (s, 3H); ^13^C-NMR (CDCl_3_, 100 MHz): δ 192.2, 157.8, 136.1, 133.3, 133.2, 132.9, 131.3, 129.2, 128.8, 128.5, 128.3, 126.4, 126.0, 125.9, 125.0, 118.9, 116.0, 56.0; MS–ESI (*m*/*z*) (relative intensity) 245.2 (70), 263.2 (M + H^+^, 100); HRMS–ESI (*m*/*z*) [M + H^+^] calcd. for C_18_H_15_ON_2_: 263.1067, found 263.1064.

4′,6-Dimethoxy-2′-methyl-[1,1′-biphenyl]-2-carbaldehyde (**5ne**). ^1^H-NMR (CDCl_3_, 400 MHz): δ 9.64 (s, 1H), 7.61 (d, *J* = 7.6 Hz, 1H), 7.45 (t, *J* = 8.4 Hz, 1H), 7.18 (d, *J* = 8.0 Hz, 1H), 7.04 (d, *J* = 8.4 Hz, 1H), 6.86 (s, 1H), 6.81 (d, *J* = 8.0 Hz, 1H), 3.85 (s, 3H), 3.79 (s, 3H), 2.05 (s, 3H), ^13^C-NMR (CDCl_3_, 100 MHz): δ 192.7, 159.4, 157.3, 138.7, 135.6, 134.4, 131.9, 128.6, 125.1, 118.9, 115.8, 115.4, 110.8, 55.9, 55.1, 20.3; MS–ESI (*m*/*z*) (relative intensity) 279.2 (M + Na^+^, 100); HRMS–ESI (*m*/*z*) [M + Na]^+^ calcd. for C_16_H_16_O_3_Na: 279.0992, found 279.0986.

2′-Formyl-6′-methoxy-2-methyl-[1,1′-biphenyl]-4-carbonitrile (**5nf**). ^1^H-NMR (CDCl_3_, 400 MHz): δ 9.61 (s, 1H) 7.65–7.61 (m, 2H), 7.57–7.51 (m, 2H), 7.25–7.24 (m, 1H), 7.23–7.22 (m, 1H), 3.78 (s, 3H), 2.09 (s, 3H); ^13^C-NMR (CDCl_3_, 100 MHz): δ 191.1, 156.6, 140.3, 139.0, 134.7, 133.2, 131.9, 131.4, 129.8, 129.2, 120.0, 118.8, 116.1, 112.1, 56.0, 19.8; MS–APCI (*m*/*z*) (relative intensity) 252.2 (M + H^+^, 100); HRMS–APCI (*m*/*z*) [M + H^+^] calcd. for C_16_H_14_O_2_N: 252.1019, found 252.1017.

## 4. Conclusions

In conclusion, we have successfully prepared a new non-C_2_-symmetric *bis*-benzimidazolium salt **1** from racemic valinol in 6 steps in good yields. The high catalytic ability of this in situ-generated Pd catalyst allowed it to expedite the Pd-catalyzed tri-*ortho*-substituted biaryl syntheses under 1 mol% Pd catalyst loading and 110 °C in 50–85% yields. In addition, this catalytic system provided, in many cases, better results than the present *bis*-NHC/Pd system for Suzuki–Miyaura cross-coupling of unactivated aryl chlorides and excellent functional group tolerance, e.g., methoxy, ketone, ester, and aldehyde groups. This method provides an alternative synthetic pathway for constructing sterically hindered biaryls.

## Data Availability

Data are contained within the article and the Appendix A.

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
