# Peer review of "Non-C2-Symmetric Bis-Benzimidazolium Salt Applied in the Synthesis of Sterically Hindered Biaryls"

_molecules, 2021, doi:10.3390/molecules26216703_

Round 1

Reviewer 1 Report

The manuscript submitted by D.-S. Lee and co-workers explores the synthesis of a non symmetric bis NHC ligand and its application in catalysis. Such type of ligands have attracted great attention for their improved performance in C-C coupling reactions. In this case, the authors have focused on their application in the Suzuki-Miyaura coupling employing aryl chlorides as starting materials. There are some precents in the literature in which similar type ligands have been used for the same purpose, however that studies have not tackled the synthesis of sterically hindered biaryls, a relevant and interesting aspect of coupling chemistry. In my opinion the manuscript is suitable for its publication in Molecules after some major revision dealing with the following points:

  • The authors state several times within the manuscript (lines 60-61 in page 2; lines 317-319 in page 10) that the isopropylgroup present in the linker of both NHC fragments is responsible for the good catalytic activity of the metal complex. In my opinion, this statement is not supported by any experimental data, for example, there is no comparation with the performance of an analogue catalyst bearing a simple ethyl linker between both NHC fragments. At first sight, the iPr group on the chain seems quite far from the reactive metal center where the reductive elimination to form the biaryl takes place.
  • Some precedents on the use of bis NHC Pd compelxes for Suzuki coupling of chloroarenes should be included ( Nan, B. Rao, M. Luo, Arkivoc 2011, 29-40), along with some important reviews closely related: a) J.-Q. Liu, X.-X. Gou, Y.-F. Han, Chem. – Asian J. 2018, 13, 2257-2276. b) M. G. Gardiner, C. C. Ho, Coord. Chem. Rev. 2018, 375, 373-388.
  • The English language needs to be polished.
  • The 1H-NMR spectra of compound 5oa is not of suitable quality for publication.

Author Response

Q1. The authors state several times within the manuscript (lines 60-61 in page 2; lines 317-319 in page 10) that the isopropyl group present in the linker of both NHC fragments is responsible for the good catalytic activity of the metal complex. In my opinion, this statement is not supported by any experimental data, for example, there is no comparation with the performance of an analogue catalyst bearing a simple ethyl linker between both NHC fragments. At first sight, the iPr group on the chain seems quite far from the reactive metal center where the reductive elimination to form the biaryl takes place.

Response: Although the iPr group on the chain might seem to be far from the reacting metal center at first sight we predict the actual distance between the two is close enough to exhibit a strong influence on the outcome of the reductive elimination in our case.

Q2. Some precedents on the use of bis NHC Pd complexes for Suzuki coupling of chloroarenes should be included (Nan, B. Rao, M. Luo, Arkivoc 2011, 29-40), along with some important reviews closely related: a) J.-Q. Liu, X.-X. Gou, Y.-F. Han, Chem. – Asian J. 201813, 2257-2276. b) M. G. Gardiner, C. C. Ho, Coord. Chem. Rev. 2018375, 373-388.

Response: These references were cited as references 31 (Nan, B. Rao, M. Luo, Arkivoc 2011, 29-40), 15 (J.-Q. Liu, X.-X. Gou, Y.-F. Han, Chem. – Asian J. 2018, 13, 2257-2276), and 16 (M. G. Gardiner, C. C. Ho, Coord. Chem. Rev. 2018, 375, 373-388). The reference number was renewed. (see revised manuscript: Molecules-1438411_1)

Q3. The English language needs to be polished.

Response: We have paid to an editing service company to help us improving the quality of English of our manuscript.

Q4. The 1H-NMR spectra of compound 5oa is not of suitable quality for publication.

Response: The 1H and 13C NMR spectra of 5oa in high quality have been provided. (see revised supporting information: molecules_SI_2)

Reviewer 2 Report

The synthesis of Bis-benzimidazolium salt and its application against the Suzuki cross-coupling reaction between aryl boronic acids and sterically hindered aryl halides has been described by Chen et al. The researchers have done a good job, however, there are a few minor changes that need to be made.

  1. Authors must mention the coordination structure of the Bis-benzimidazolium salt with palladium in the manuscript.
  2. The compound (5oc) is not found in the manuscript as mentioned in line no. 116.
  3. The reaction conditions are repeatedly mentioned over the arrows and below the tables.
  4. Mass analysis and elemental analysis are necessary. The authors gave only HRMS analysis for compounds S4, 1,2, 5nb, 5nf, 5ne.
  5. Line no.71 salts should be salt.
  6. At the end of tables No. 1,2 & 3. 

Author Response

Q1. Authors must mention the coordination structure of the Bis-benzimidazolium salt with palladium in the manuscript.

Response: We didn't have direct X-ray structure evidence to show that salt 1 formed a complex with Pd since we didn't obtain a crystal of it. However, the formation of similar complexes has been reported in the literature, e.g. J. De Tovar, F. Rataboul, L. Djakovitch, ChemCatChem 2020, 12, 5797. In addition, the control test in table 1 (entry 22) indicates strongly that an effective catalytic system did form in the reaction. Supported by these results, we proposed that a seven-membered complex formed in the reaction process.

Q2. The compound (5oc) is not found in the manuscript as mentioned in line no. 116.

Response: The compound number 5oc was replaced with 5nc (Line 115). (see revised manuscript: Molecules-1438411_1)

Q3. The reaction conditions are repeatedly mentioned over the arrows and below the tables.

Response: It has been revised in Scheme 1. (see revised manuscript: Molecules-1438411_1)

Q4. Mass analysis and elemental analysis are necessary. The authors gave only HRMS analysis for compounds S4, 1,2, 5nb, 5nf, 5ne.

Response: The mass analyses of compounds S4, 1, 2, 5nb, 5nf, and 5ne have been added. (see revised manuscript: Molecules-1438411_1)

Q5. Line no.71 salts should be salt.

Response: The mistake in Scheme 1 has been revised. (see revised manuscript: Molecules-1438411_1)

Q6. At the end of tables No. 1,2 & 3. 

Response: We didn't understand the meaning of this comment.

Round 2

Reviewer 1 Report

The authors have made some corrections required in the the first round of comments. However, the question Q1 has not been satisfactorily answered. 

The authors state in their answer  "we predict the actual distance between the two is close enough to exhibit a strong influence on the outcome...". As far as I am aware, predictions are not a valuable tool in rigorous scientific works. 

In the conclusions of the paper (page 10 of the manuscript)., the authors affirmed categorically: 

"The steric hindrance of the isopropyl group on the non-C2-symmetric bis-benzimidazolium salt 1 did effectively enhance catalytic capability of the system formed by this ligand and Pd(dba)2." 

This sentence can only be kept in the conclusions if the authors demonstrate it experimentally, as mentioned in the first round of the revision process. 

If the authors are unable to perform such demonstration, then they must remove the sentence "The steric hindrance of the isopropyl group on the non-C2-symmetric bis-benzimidazolium salt 1 did effectively enhance catalytic capability of the system formed by this ligand and Pd(dba)2." from the conclusions, since it is only based on personal predictions, not in scientific proofs.

Author Response

Response: We decided to remove the sentence “The steric hindrance of the isopropyl group on the non-C2-symmetric bis-benzimidazolium salt 1 did effectively enhance catalytic capability of the system formed by this ligand and Pd(dba)2.” from conclusion. Therefore, the manuscript has been revised. (see revised manuscript: Molecules-1438411_2)